# Immunization with *centrin*-Deficient *Leishmania braziliensis* Does Not Protect against Homologous Challenge

**DOI:** 10.3390/vaccines12030310

**Published:** 2024-03-15

**Authors:** Francys Avendaño-Rangel, Gabriela Agra-Duarte, Pedro B. Borba, Valdomiro Moitinho, Leslye T. Avila, Larissa O. da Silva, Sayonara M. Viana, Rohit Sharma, Sreenivas Gannavaram, Hira L. Nakhasi, Camila I. de Oliveira

**Affiliations:** 1Instituto Gonçalo Moniz, FIOCRUZ, Salvador 40296-710, BA, Brazil; francysav.rangel@gmail.com (F.A.-R.); gabriela.agra@fiocruz.br (G.A.-D.); pbriito09@gmail.com (P.B.B.); valdomiro.moitinho@fiocruz.br (V.M.); leslye.avila@fiocruz.br (L.T.A.); larissasilva20.2@bahiana.edu.br (L.O.d.S.); sayomelo@gmail.com (S.M.V.); rohitah.sharma@gmail.com (R.S.); 2Programa de Pós-Graduação em Ciências da Saúde, Faculdade de Medicina da Bahia, Universidade Federal da Bahia, Salvador 40170-110, BA, Brazil; 3Division of Emerging and Transfusion Transmitted Diseases, CBER, FDA, Silver Spring, MD 20993, USA; sreenivas.gannavaram@fda.hhs.gov (S.G.); hira.nakhasi@fda.hhs.gov (H.L.N.); 4Instituto Nacional de Ciência e Tecnologia em Doenças Tropicais (INCT-DT), Salvador 40296-710, BA, Brazil

**Keywords:** live attenuated vaccine, immunization, vaccination, cutaneous leishmaniasis

## Abstract

Immunization with various *Leishmania* species lacking *centrin* induces robust immunity against a homologous and heterologous virulent challenge, making *centrin* mutants a putative candidate for a leishmaniasis vaccine. Centrin is a calcium-binding cytoskeletal protein involved in centrosome duplication in higher eukaryotes and *Leishmania* spp. lacking centrin are unable to replicate *in vivo* and are non-pathogenic. We developed a *centrin*-deficient *Leishmania braziliensis* (*LbCen^−/−^*) cell line and confirmed its impaired survival following phagocytosis by macrophages. Upon experimental inoculation into BALB/c mice, *LbCen^−/−^* failed to induce lesions and parasites were rapidly eliminated. The immune response following inoculation with *LbCen^−/−^* was characterized by a mixed IFN-γ, IL-4, and IL-10 response and did not confer protection against *L. braziliensis* infection, distinct from *L. major*, *L. donovani*, and *L mexicana* centrin-deficient mutants. A prime-boost strategy also did not lead to a protective immune response against homologous challenge. On the contrary, immunization with *centrin*-deficient *L. donovani* (*LdonCen^−/−^*) cross-protected against *L. braziliensis* challenge, illustrating the ability of *LdonCen^−/−^* to induce the Th1-dominant protective immunity needed for leishmaniasis control. In conclusion, while *centrin* deficiency in *L. braziliensis* causes attenuation of virulence, and disrupts the ability to cause disease, it fails to stimulate a protective immune response.

## 1. Introduction

Leishmaniasis is a parasitic disease caused by protozoans of the genus *Leishmania* that are transmitted by sand fly insects. There are several clinical presentations of leishmaniasis which vary according to the parasite species. The leishmaniases cause high morbidity and mortality worldwide and are an important public health problem [1]. In 2021, Brazil reported 15,023 cases of cutaneous/mucosal leishmaniasis [2] caused mainly by *Leishmania braziliensis*. Although different vaccine formulations have conferred protection against *Leishmania* spp. in animal models [3], to this date, there are no leishmaniasis vaccines approved for human use.

*Leishmania* infection can stimulate a protective response that depends upon the generation of IFN-γ-producing CD4^+^ T cells, leading to macrophage activation and subsequent parasite killing [4]. On the other hand, disease development is mainly driven by the production of non-protective Th2-associated cytokines IL-4, IL-10, IL-13, and TGF-β [5,6]. Durable protective immunity acquired following leishmanization led to the development of several live attenuated *Leishmania* strains as prophylactic vaccines [7]. The possibility of using genetically modified *Leishmania* spp. as candidate vaccines has expanded given the feasibility of large-scale and cost-effective production [8]. Adaptation of CRISPR/Cas9 methods to *Leishmania* genome editing enabled the development of marker-free *Leishmania* strains [9]. Research towards the deletion of genes related to parasite survival and/or virulence is promising, as these modifications maintain the antigenic repertoire of wild-type counterparts and thus the ability to induce a broad-spectrum immune response while eliminating the pathology associated with infection [10]. Centrin 1 is a calcium-binding protein that plays a fundamental role in centrosome duplication [11]. Centrin 1 was initially characterized in *L. donovani* as a homolog of human *centrin* 2, and infection of mice with centrin-deficient *L. donovani* (*LdCen^−/−^*) induces a multifunctional T-cell response that significantly reduces the parasite load upon virulent challenge, conferring protection against disease [12].

*L. braziliensis* has been little explored in terms of vaccine development despite its importance as the causative agent of mucosal and disseminated leishmaniasis, two severe clinical manifestations of leishmaniasis [13]. Given the promising results obtained with centrin-deficient *Leishmania* spp., we developed a *centrin*-deficient *L. braziliensis* cell line (*LbCen*^−/−^) [14]. We previously showed that *centrin* deletion did not cause off-target mutations and that *LbCen^−/−^* axenic amastigotes become multinucleated cells and display impaired survival in macrophages similar to other centrin-deleted *Leishmania* species [12,15,16]. *In vivo*, *LbCen^−/−^* failed to induce cutaneous lesions in BALB/c mice [14]. Herein, we explored *LbCen^−/−^* as a vaccine candidate against cutaneous leishmaniasis caused by *L. braziliensis*. Although *LbCen^−/−^* is safe and non-pathogenic, it did not induce a protective immune response, differently from what was observed in mice that healed from a primary infection with *Leishmania braziliensis* wild type *(LbWT)* or mice inoculated with centrin-deficient *L. donovani (LdonCen^−/−^)*. Our results point out the differences in the ability of attenuated *Leishmania* spp. to induce protective immune responses, as well as the divergent mechanisms of protection that may be necessary, collectively highlighting the hurdles of vaccine development against some species causing American Tegumentary Leishmaniasis.

## 2. Materials and Methods

### 2.1. Parasite Culture

*L. braziliensis* wild-type (*LbWT*) parasites (strain MHOM/BR/01/BA788) were grown in Schneider’s insect medium (Sigma Aldrich) supplemented with 20% heat-inactivated FBS, 2 mM of glutamine, 100 U/mL of penicillin, and 100 mg/mL of streptomycin (all from Thermo Scientific) at 26 °C. Centrin-deficient *L. braziliensis* (*LbCen^−/−^*) [16] was maintained in Schneider’s insect medium (Sigma Aldrich) supplemented with 20% heat-inactivated FBS, 2 mM of glutamine, 100 U/mL of penicillin, and 100 mg/mL of streptomycin (all from Thermo Scientific) and supplemented with neomycin (50 µg/mL) (Promega) and puromycin (10 µg/mL) (Sigma).

### 2.2. Bone Marrow-Derived Macrophage (BMDM) Culture and In Vitro Exposure to LbCen^−/−^

Bone marrow cells were isolated from femurs and tibias of BALB/c mice and cultured with RPMI 1640 medium (Sigma-Aldrich) supplemented with 10% fetal bovine serum (FBS), 100 U/mL of penicillin, 100 mg/mL of streptomycin, and 2.5% HEPES (all from Invitrogen) and further complemented with conditioned media from the mouse fibroblast cell line L929 (ThermoFisher). After differentiation, bone marrow-derived macrophages (BMDMs) were seeded at 3 × 10^5^ cells/500 μL/coverslip per well in 24-well plates. BMDM monolayers were washed to remove non-adherent cells and infected with stationary-phase promastigotes *LbWT* or *LbCen*^−/−^ at a 10:1 parasite/host cell ratio in supplemented RPMI 1640/well. Plates were incubated at 37 °C under 5% CO_2_ for 6 h or 24 h. Coverslips were extensively washed to remove any non-internalized *Leishmania* and were methanol-fixed and stained with hematoxylin and eosin (H&E). *Leishmania* uptake and internalized parasites were assessed by observing 200 macrophages, and the number of cells with and without *Leishmania* as well as the total number of intracellular parasites were counted by optical microscopy (magnification ×1000). Alternatively, coverslips were washed after 24 h of infection and further cultured in 500 μL of supplemented RPMI 1640/well at 37 °C and 5% CO_2_ for a total of 72 h. Then, cells were washed and cultured in supplemented Schneider’s insect medium for 5 days at 26 °C. Promastigote numbers were determined at days 3 and 5 by counting in a hemocytometer.

### 2.3. Nitric Oxide and Cytokine Quantification

BMDMs were primed or not with lipopolysaccharide (LPS) (100 ng/mL) (Invitrogen) and infected with *LbWT* or *LbCen*^−/−^ as described above for 24 h. Cell culture supernatants were collected after 24 h for nitric oxide quantification using a Griess reagent [17]. Cytokine levels (IL-10, TNF-a, IL-12p70, and IL-1B) were determined by sandwich ELISA, following the manufacturer’s instructions (Invitrogen).

### 2.4. Safety and Efficacy of LbCen^−/−^ in BALB/c Mice

BALB/c mice (n = 10) were inoculated in the right ear dermis with *LbCen^−/−^* (3 × 10^6^ stationary-phase promastigotes), using a 27G needle. Ear thickness was measured weekly using a digital caliper (Thermo Scientific). Parasite load was evaluated at the inoculation site (ear) and in draining lymph nodes (dLNs), two and five weeks later (five mice per time point), by limiting dilution analysis, as previously described [18].

To evaluate protection, mice (n = 12) were inoculated with *LbCen^−/−^*, as described above. Five weeks later, mice were challenged in the left ear dermis with 10^5^ *LbWT*. Controls consisted of age-matched naïve mice (n = 10) infected with 10^5^ *LbWT*. Parasite load was determined five weeks after challenge (6 mice per time point), as described above.

### 2.5. Leishmanization

BALB/c mice (n = 12) were inoculated in the right ear dermis with *LbWT* (10^5^ stationary-phase promastigotes), using a 27G needle. Ear thickness was measured weekly using a digital caliper (Thermo Scientific). Ten weeks later, after lesion healing, we evaluated the cellular and humoral response. Delayed-Type Hypersensitivity (DTH) was evaluated (n = 6 mice) by inoculation of *L. braziliensis* SLA (10 ug/mL), prepared as described elsewhere [19], in the left ear dermis. Ear induration was measured 24, 48, and 72 h later, using a digital caliper (Thermo Scientific). For the humoral response (n = 6 mice), MaxiSorp 96-plates were coated with Soluble Leishmania Antigen (SLA) (10 ug/mL) and incubated overnight at 4 °C. Plates were washed with PBS 1X +0.05% Tween 20 (PBS-T) and blocked with 5% non-fat milk in PBS-T for 2 h at room temperature. Sera were diluted 1/100 in PBS-T +5% non-fat milk and were incubated overnight at 4 °C. The plates were washed, incubated with IgG1 or IgG2a HRP-conjugated antibodies (Invitrogen) (1/1000 in PBS-T +5% non-fat milk) for 2 h at room temperature. The plates were washed, TMB solution (Invitrogen) was added, and reactions were stopped with H_2_SO_4_ 2N. The plates were read at 450 nm using a FilterMax F3 Multi-mode Microplate Reader (Molecular Devices). SoftMax Pro Software (Version 6.3) was used to obtain an indirect measure of antibody titers in serum samples. Sera from naïve mice were used as controls.

Ten weeks after the primary *LbWT* infection, healed (leishmanized) mice (n = 6) were challenged with 10^5^ stationary-phase *LbWT* promastigotes, inoculated in the left ear dermis. Controls consisted of age-matched naïve mice (n = 6) inoculated with stationary-phase *LbWT* promastigotes. Parasite load was determined five weeks after challenge, as described above, with 6 mice per time point. To evaluate the cellular response, dLN cells (1 × 10^6^) were seeded on 24 well-plates and stimulated with SLA (10 ug/mL) or ConA (1 ug/mL) or left unstimulated, in supplemented RPMI. Cells were incubated at 37 °C and 5% CO_2_ for 48 h and cytokine production was assayed in culture supernatants by a Luminex Assay (MILLIPLEX MAP Mouse Cytokine/Chemokine Magnetic Bead Panel, Merck Millipore, Cat#MCYTOMAG-70K), following the manufacturer’s instructions. Data acquisition was performed with the Luminex LX200 (Merck Millipore). Mean Fluorescent Intensity was calculated against standards using the Milliplex Analyst software Version 5.1.

### 2.6. Immunization with LbCen^−/−^ and Challenge with LbWT

BALB/c mice (n = 6) were inoculated in the right ear dermis with *LbCen^−/−^* (3 × 10^6^ stationary-phase promastigotes) or *LbWT* (10^5^ stationary-phase promastigotes) using a 27G needle. Ear thickness was measured weekly using a digital caliper (Thermo Scientific). The cellular response (DTH, n = 3 mice; re-stimulation of dLN cells with SLA, n = 3 mice) and humoral immune response (IgG1 and IgG2a, n = 6 mice) were evaluated as described above at the following time points: ten weeks after the healing of the primary infection with *LbWT* (leishmanization) and four weeks after inoculation with *LbCen^−/−^*.

### 2.7. Efficacy of Immunization with LbCen^−/−^ versus Leishmanization

BALB/c mice (n = 6 per group) were either leishmanized or immunized with *LbCen^−/−^* as described above. Ten weeks after leishmanization and six weeks after immunization, mice were challenged with *LbWT* (10^5^ stationary-phase promastigotes). Controls consisted of age-matched naïve mice (n = 6) inoculated with stationary-phase *LbWT* promastigotes. Six weeks post challenge, mice were euthanized for the determination of parasite load and cytokine production in dLN cells. Challenged ears were excised and fixed in 10% formalin. Tissues were embedded in paraffin, cut, and stained with hematoxylin and eosin (H&E). Slides were examined under a light microscope and images were obtained using a Slide scanner Olympus VS110 system and photographs were obtained using OlyVIA Software (version 3.8).

### 2.8. Primary and Booster Immunization

BALB/c mice (n = 8) were inoculated in the right ear dermis with *LbCen^−/−^* (3 × 10^6^ stationary-phase promastigotes), and two weeks later, the mice were boosted at the same site. Two weeks after the booster inoculation, the cellular response (DTH and re-stimulation of dLN cells with SLA) was evaluated as described above. The frequency of central memory T cells (TCMs) was determined by flow cytometry. Briefly, dLN cells (5 × 10^5^) were incubated with Fc Block (BD Biosciences, clone 2.4G2) for 5 min at 4 °C and then stained with CD4 (eBioscience, clone GK1.5), CD44 (BD Pharmingen), and CD62L (BD Pharmingen) antibodies in the dark for 20 min at 4 °C. The cells were washed in PBS and resuspended in PBS containing Hoechst 33258 at 5 ug/mL. Acquisition was performed using an LSRFortessa™ flow cytometer (BD Biosciences) and analyzed using FlowJo Software version 10. Isotype controls for each antibody used under similar conditions indicated specific binding of the test antibody. Electronic compensation was performed with single-stained cells with individual mAbs used in the test samples.

To evaluate the efficacy of the prime-boost strategy, mice (n = 8) were immunized as described and, five weeks later, were challenged with *LbWT* (10^5^ stationary-phase promastigotes). Controls consisted of age-matched naïve mice (n = 8) inoculated with stationary-phase *LbWT* promastigotes. Lesion development was measured weekly. Five weeks after the challenge, parasite load, cytokine response in dLN cells stimulated with SLA, and the frequency of TCM were determined, as described above.

### 2.9. Cross-Protection: Immunization with LdonCen^−/−^ and Challenge with LbWT

BALB/c mice (n = 6) were inoculated in the right ear dermis with 3 × 10^6^ stationary-phase *LbCen^−/−^* or with metacyclic *Ldon*Cen^−/−^ via the tail vein. Five weeks later, mice were challenged with *LbWT* (10^5^ stationary-phase promastigotes) in the left ear dermis. Controls consisted of age-matched naïve mice (n = 6). Ear thickness was measured weekly. Six weeks post challenge, parasite load was determined by limiting-dilution analysis, as described above.

### 2.10. Statistical Analysis

The course of disease was plotted individually for mice in all experimental and control groups. Comparisons between two groups were performed using Mann–Whitney non-parametric *t*-tests and comparisons among more than two groups were performed using the Kruskal–Wallis method. Data are presented as the mean ± error of the mean. Analyses were conducted using Prism (V.8.0, GraphPad) and a *p*-value ≤ 0.05 was considered significant.

## 3. Results

### 3.1. LbCen^−/−^ Parasites Exhibit Impaired Growth and Survival in BMDMs

BMDMs were infected with *LbCen^−/−^* or *LbWT* and, after 24 h and 72 h, the number of infected macrophages and intracellular parasites was determined. Exposure to *LbCen^−/−^* led to significantly reduced survival (*p* < 0.05) compared to *LbWT*, both at 24 h and 72 h (Figure 1A,B). After 72 h, the BMDMs were washed and cultured in a supplemented Schneider medium to determine the number of recovered parasites. At both time points, we observed a significantly decreased (*p* < 0.05) recovery of *LbCen^−/−^* compared to *LbWT* (Figure 1C). This outcome was not due to impaired uptake by the host cell, as the number of BMDMs harboring *LbCen^−/−^* or *LbWT* was similar after 6 h of exposure (Appendix A) as was the number of intracellular amastigotes. These results confirm that lack of *centrin* in *L. braziliensis* impairs survival and replication within the host cell.

### 3.2. Macrophage Exposure to LbCen^−/−^ or LbWT Induces Similar Cytokine Response

BMDMs were exposed to *LbCen^−/−^* or *LbWT* in the presence or absence of LPS, and cytokine production was evaluated after 24 h. Levels of TNF were similar in both *LbCen^−/−^* and *LbWT* infections and significantly higher (*p* < 0.05) compared to controls (Figure 2A). Levels of IL-12p70 were also similar for *LbCen^−/−^* and *LbWT* and significantly lower (*p* < 0.05) compared to controls stimulated with LPS (Figure 2B). Exposure to *LbWT* impaired IL-1β secretion, whereas *LbCen^−/−^* did not (Figure 2C). IL-10 levels (Figure 2D) and NO production were not changed in comparison to controls (Figure 2E). Taken together, these results show that the macrophages exposed to *LbWT* or *LbCen^−/−^* parasites produce a similar innate immune response, except for IL-1β.

### 3.3. Immunization with LbCen^−/−^ Does Not Cause Lesions and Is Safe

The deletion of *centrin* affects amastigote growth, and mice inoculated with *centrin*-deficient *Leishmania* do not develop disease [15,16]. Deletion of the *centrin* in *L. braziliensis* reproduced this avirulent phenotype, as BALB/c inoculated with *LbCen^−/−^* parasites failed to develop cutaneous lesions (Figure 3A), whereas inoculation of *LbWT* leads to lesions that develop and heal spontaneously [13]. Two weeks after inoculation, the parasite load was significantly lower (*p* < 0.05) in the ears of mice inoculated with *LbCen^−/−^* compared to *LbWT* (Figure 3B). Viable parasites were recovered from the inoculation site for both *LbWT* and *LbCen^−/−^* groups at 2 weeks p.i., but only from the *LbWT* group and not from the *LbCen^−/−^* group from the dLNs 2 weeks p.i. (Figure 3B). Five weeks post inoculation, *LbCen^−/−^* parasites were not detected at either the inoculation site or in dLNs (Figure 3C), whereas in *LbWT*-infected mice, parasites were detected at both sites.

### 3.4. Immunization with LbCen^−/−^ Does Not Protect against LbWT Infection

To evaluate the protective capacity of *LbCen^−/−^*, we immunized BALB/c mice with *LbCen^−/−^* and, 5 weeks later, challenged them with *LbWT*. Surprisingly, immunization with *LbCen^−/−^* did not protect against *LbWT* infection (Figure 4A). Mice developed cutaneous lesions that were comparable to those observed in naïve controls upon a challenge infection with *LbWT* at week 10 (Figure 4A). In a separate experiment, after 5 weeks of immunization followed by five weeks of WT challenge, there was no difference in the parasite load in the two groups, both at the inoculation site and in dLNs (Figure 4B and Figure 4C, respectively). Thus, a lack of centrin in *L. braziliensis* renders parasites unable to cause pathology but incapable of conferring protection against infection with wild-type *L. braziliensis*, differently from the centrin deletion mutation of other *Leishmania* spp. [12,15,16].

### 3.5. Immunization with LbCen^−/−^ Cannot Recapitulate the Immune Response Induced by a Healed Primary Infection with LbWT

Leishmanization, known as the deliberate inoculation of *Leishmania* parasites to provide protective immunity to re-infection, is effective and has been successfully tested in pre-clinical animal models [20,21] and in humans [22]. We thus evaluated the immune response in mice that healed from a primary infection with *LbWT*. Following inoculation of *LbWT*, BALB/c mice developed lesions that peaked at around week 5 and spontaneously healed by week 10 (Appendix A). These primary infected and healed mice mounted a DTH response against SLA (Appendix A) and produced both IgG1 and IgG2a (Appendix A). Upon re-infection of healed mice with *LbWT*, lesions did not develop (Appendix A), differently from the control (naïve) mice. This protective outcome is accompanied by a significantly lower (*p* < 0.01) parasite load at the inoculation site but not in dLNs (Appendix A). Mice that healed from a primary infection also produced significantly (*p* < 0.01) lower levels of IFN-γ, IL-4, and IL-10 (*p* < 0.01) (Appendix A) compared to control (naïve) mice. Thus, experimental infection with *LbWT* induces disease development that is spontaneously cured in BALB/c mice, and healed mice are immune to challenge.

Next, we wanted to compare the immune response of leishmanized versus immunized animals. As observed before, *LbCen*^−/−^ does not cause lesion development, differently from *LbWT* (Figure 5A). Although the DTH responses in the two groups were not significantly different (Figure 5B), IgG1 and IgG2a levels (Figure 5C) were significantly higher (*p* < 0.01) in leishmanized mice compared to immunized animals. Moreover, while IFN-γ production was similar in the two groups, IL-4, IL-10 and IL-17 levels were higher in mice immunized with *LbCen*^−/−^ (Figure 5D). These results showed that the immune responses that developed upon inoculation with *LbCen*^−/−^ were not Th1-dominant, contrary to that developed by mice that healed from a primary infection with *LbWT*.

Given the immune responses developed by mice immunized with *LbCen*^−/−^ versus leishmanized mice, we compared the outcome of a challenge with *LbWT* in these two settings. At the peak of lesion development, tissue sections from the ears of *LbCen^−/−^*-immunized mice showed signs of chronic inflammation, features not observed in leishmanized mice (Figure 6A), yet not as prominent as those observed in the naïve mice. On the contrary, leishmanized mice showed only mild signs of inflammation. Leishmanized mice presented a significantly lower (*p* < 0.01) disease burden (AUC) compared to mice inoculated with *LbCen^−/−^* or control (naïve) mice (Figure 6B) and significantly lower (*p* < 0.01) parasite load at the inoculation site, compared to *LbCen^−/−^*-inoculated or control (naïve) mice (Figure 7C). While the three experimental groups displayed similar levels of IFN-γ, detected upon stimulation of dLN cells (Figure 6D), the production of IL-4, IL-10, and IL-17 was significantly higher in mice inoculated with *LbCen^−/−^* (Figure 6D). Thus, we can speculate that the cellular recruitment induced by immunization with *LbCen^−/−^* may have favored parasite establishment. Taken together, these results demonstrate that inoculation with *LbCen^−/−^* cannot recapitulate the immune profile induced by leishmanization and, as such, is incapable of conferring protection against challenge with *L. braziliensis*.

### 3.6. Priming/Boosting Induces Prolonged Parasite Persistence

For most vaccine strategies, multiple immunizations are required to induce efficient protection [23]. Although this has not been necessary in studies performed with *Leishmania* spp. lacking centrin [15,16], we investigated whether a second immunization with *LbCen^−/−^* could boost the primary immune response and, hence, alter the outcome of infection after challenge. Priming and boosting with *LbCen^−/−^* parasites continued to be safe as mice remained lesion-free (data not shown). The cellular response, measured by DTH (Figure 7A), was similar to that observed with a single immunization (Figure 5B). The recall response also showed that while IFN-γ was produced, IL-10 was detected at comparatively higher levels (Figure 7B). Two immunizations with *LbCen^−/−^* were also not capable of generating a significant increase in CD4^+^ T central memory cells (CD4^+^CD44^+^CD62L^+^) (Figure 7C), as seen upon a single immunization with *LmexCen^−/−^* [16]. Given that a booster inoculation with *LbCen^−/−^* did not alter the immune response observed earlier, we expected that the outcome of a challenge infection would also be the same. Indeed, mice receiving two inoculations with *LbCen^−/−^* continued to be susceptible to a challenge infection with *LbWT* (Figure 7D), with lesions developing similarly to the control, naïve mice. The parasite load in the two groups was similar (Figure 7E). Lastly, no significant differences were observed in the production of IFN-γ (Figure 7F) or IL-10 (Figure 7G), comparing mice primed and boosted with *LbCen^−/−^* and control, naïve mice. Collectively, our results show that a booster *LbCen^−/−^* immunization does not alter the lack of protection observed upon a single immunization.

### 3.7. Immunization with LdonCen^−/−^ Cross-Protects against LbWT

Previously, it was shown that immunization with *LdonCen^−/−^* induces cross-protection against *L. braziliensis* injected subcutaneously [12]. We sought to investigate whether a similar outcome would be observed using the *L. braziliensis* strain employed in the generation of our centrin-deleted mutant, using an intradermal model of infection. We thus compared the outcome of immunization with *LdonCen^−/−^* versus *LbCen^−/−^* followed by a challenge with *LbWT* in the ear dermis. We confirmed that immunization with *LdonCen^−/−^* protects against intradermal inoculation of *LbWT*, differently from *LbCen^−/−^* (Figure 8A). Protection was paralleled by a significantly lower parasite (*p* < 0.01) load at the inoculation site and in dLNs (Figure 8B), suggesting that the immune response induced by *LdonCen^−/−^* controls parasite load even at distal sites.

## 4. Discussion

The current therapeutic options for leishmaniasis remain limited because of toxicity, drug administration challenges, treatment failure, and drug resistance [24]. Protection against leishmaniasis can be achieved through leishmanization [25], but this method presents safety concerns. To this end, attenuated *Leishmania* spp. cell lines are an attractive alternative, as safety issues can be overcome while the delivery of the near-complete antigen repertoire of the parasite is achieved. Indeed, attenuated *Leishmania* lacking the *centrin 1* gene is being intensively explored in the context of vaccination against visceral [12] and cutaneous leishmaniasis [15,16]. Herein, we explored this possibility by using a *centrin-1*-deficient *L. braziliensis* [14], aimed at the development of a vaccine against American Tegumentary Leishmaniasis.

While deletion of the *centrin 1* gene in *L. braziliensis* did not alter the ability of promastigotes to successfully enter the host cell, intracellular *LbCen^−/−^* proliferation was significantly reduced. In addition, exposure of macrophages to either *LbCen^−/−^* or *LbWT* led to similar outcomes regarding the production of innate mediators such as TNF, IL-12, IL-10, or NO. Infection of dendritic cells with *L. braziliensis* resulted in elevated IL-12 production [26], an effect also observed in vivo [27]. Also in vivo, inflammasome activation was shown to be involved in restricting *Leishmania* growth [28]. Herein, IL-1β was significantly increased upon infection with *LbCen^−/−^* compared to *LbWT*. We can speculate that *LbCen^−/−^* may have induced inflammasome activation and, hence, reduced parasite replication.

Inoculation of BALB/c mice with *LbCen^−/−^* was deemed safe as parasites did not induce lesion development, corroborating results obtained with *centrin-1*-deficient *Leishmania* spp. We next evaluated its ability to confer protection against *LbWT*. To our surprise, immunization with *LbCen^−/−^* failed to inhibit lesion development following challenge with live *LbWT*, contrary to what has been published for counterparts in the *Leishmania* subgenus, *L. donovani* [12], *L. major* [15], and *L. mexicana* [16]. Importantly, protection has been achieved even in the context of sand-fly-transmitted parasites [15], considered the gold standard of challenge model in leishmaniasis.

Protective immunity against *L. major* requires low levels of the persisting antigen [29,30]. Herein, *LbCen^−/−^* parasites were detected two weeks after intradermal inoculation into BALB/c mice but were rapidly cleared and could not be retrieved at a later time point (five weeks). These results differ from those reported for *LdonCen^−/−^* [12], *LmajorCen^−/−^* [15], and *LmexCen^−/−^* [16], the latter also employed in intradermal infections, in which centrin-deficient parasites were detected five weeks post inoculation, indicating their enhanced ability to survive in the vertebrate host, compared to *LbCen^−/−^*. Data from immunosuppressed mouse models showed persistence of *LmCen^−/−^* parasites 15 weeks post inoculation, suggesting that a low level of parasites may be important for acquiring protective immunity [15]. Moreover, mice primed and boosted with *LbCen^−/−^* continued to show susceptibility to infection, indicating that *centrin*-deficient *L. braziliensis* is an extremely attenuated strain, leading to rapid clearance by BALB/c mice.

We showed that BALB/c mice that heal from a primary infection with *LbWT* develop protective immunity against re-infection, analogous to leishmanization models. In this experimental demonstration of leishmanization, immunity was associated with elevated IgG1, IG2a levels, and IFN-γ production by dLN cells, all paralleled by a significantly decreased parasite load in challenged mice. In mice inoculated with *LbCen^−/−^*, an opposite result was observed: mice mounted a Th2-biased immune response, with elevated levels IL-4, IL-10, and IL-17 and lower levels of IgG1 and IgG2a antibodies. Although IFN-γ production was similar comparing *LbCen^−/−^*-immunized and leishmanized mice, IL-4 and IL-10 were only upregulated in mice inoculated with *LbCen^−/−^*. In *L. major*-infected BALB/c mice, the early production of IL-4 drives a Th2 response and susceptibility to disease, and treatment with anti-IL-4 or the use of IL-4-deficient mice abrogates this effect [31,32]. In C57BL/6 mice that heal from a primary infection with *L. major*, IL-10 is necessary for the long-term parasite persistence, and a sterile cure can be achieved in mice lacking both IL-4 and IL-10 [33]. Interestingly, immunity conferred by *LmexCen^−/−^* was mediated mainly by a reduction in IL-4 and IL-10 levels rather than an increase in IFN-γ, thus enabling the control of infection and inhibiting disease development [16]. Herein, in the same genetic background (BALB/c), immunization with *LbCen^−/−^* induced an opposing effect: upregulation in IL-4, IL-10, and IL-17. In comparison to *L. major*, *L. braziliensis* induces significantly lower expression of IL-4, IL-10, and IL-13 in the early phase of infection in BALB/c mice [27]. In C57BL/6 mice infected with *LbWT*, authors observed an expansion of IFN-γ-, IL-10-, and IL-17-producing CD4^+^ T cells at the time when lesions peaked (four weeks). By eight weeks post infection, when lesions healed, the frequency of these populations decreased, suggesting that the fine balance between pro-inflammatory and regulatory cytokines is linked to controlled *L. braziliensis* infection [26]. In C57BL/6 mice infected with *L. infantum*, IL-17A and IFN-γ potentiated NO production, restricting parasite growth in macrophages [34]. We thus suggest that although IFN-γ is induced by immunization with *LbCen^−/−^*, the unabated IL-4 and IL-10 production accompanied by sustained IL-17 are associated with a lack of protection.

## 5. Conclusions

We highlight that the deletion of *centrin* in *L. braziliensis* replicated the phenotype observed in *L. mexicana, L. major*, and *L. donovani*, with the major difference that a mixed immune response, with elevated IL-4 and IL-10 production, was observed. This cannot be attributed to the presence of Leishmania RNA Virus (LRV) [35], as the strain of *L. braziliensis* used in this study does not carry LRV (unpublished). While immunization with *LmexCen^−/−^* promotes changes in the host cell leading to M1 polarization [36], *LmajorCen^−/−^* modulates tryptophan metabolism, promoting the production of IL-12 rather than IL-10 and TGF-β [37]. It remains to be investigated if the metabolic immune regulation in *LbCen^−/−^* infection disfavors the replication of parasites, rapidly clearing the infection. Alternatively, other innate immune responses observed in *LdonCen^−/−^* may fail to materialize in *LbCen^−/−^*, resulting in a lack of protection. The divergent mechanisms of protection induced by *LmajorCen^−/−^* and *LmexCen^−/−^* highlight the need for fine tuning vaccination strategies to control American Tegumentary Leishmaniasis. It remains to be tested if *LmajorCen^−/−^*, a dermotropic strain, can confer protection against *L. braziliensis* challenge. Such studies will illustrate the common correlates of protection that could be targeted in a pan-leishmania vaccine.

## Figures and Tables

**Figure 1 vaccines-12-00310-f001:**
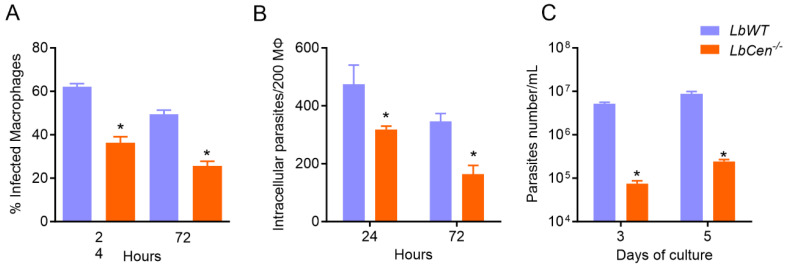
Impaired survival of *LbCen^−/−^* in murine macrophages. Bone marrow-derived macrophages (BMDMs) were infected with *LbWT* or *LbCen^−/−^* promastigotes. After 24 h, cells were washed and stained with H&E (at 24 h or 72 h). The percentage of infection (**A**) and the number of amastigotes per 200 macrophages (**B**) were determined by optical microscopy. Alternatively, infected macrophages were washed (72 h) and cultured in Schneider medium. Three or five days later, the number of promastigotes was evaluated (**C**). Results are presented as means ± SEM and are pooled from two independent experiments, each performed in quadruplicate. * *p* < 0.05.

**Figure 2 vaccines-12-00310-f002:**
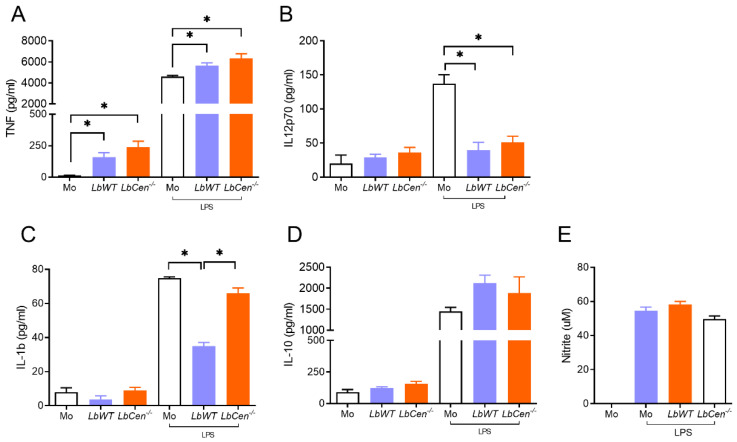
Innate immune response in *LbCen^−/−^*-infected macrophages. Bone marrow-derived macrophages (BMDMs) were primed with LPS (100 ng/mL) and were later infected with *LbWT or LbCen^−/−^* promastigotes. Controls (Mo) were left unprimed. After 24 h, culture supernatants were collected and the presence of TNF (**A**), IL-10 (**B**), IL-1B (**C**), and IL-10 (**D**) was determined by ELISA. (**E**) Nitrite concentration was evaluated by Griess reaction. Results are presented as mean ± SEM and are from a representative experiment, performed in quadruplicate. * *p* < 0.05.

**Figure 3 vaccines-12-00310-f003:**
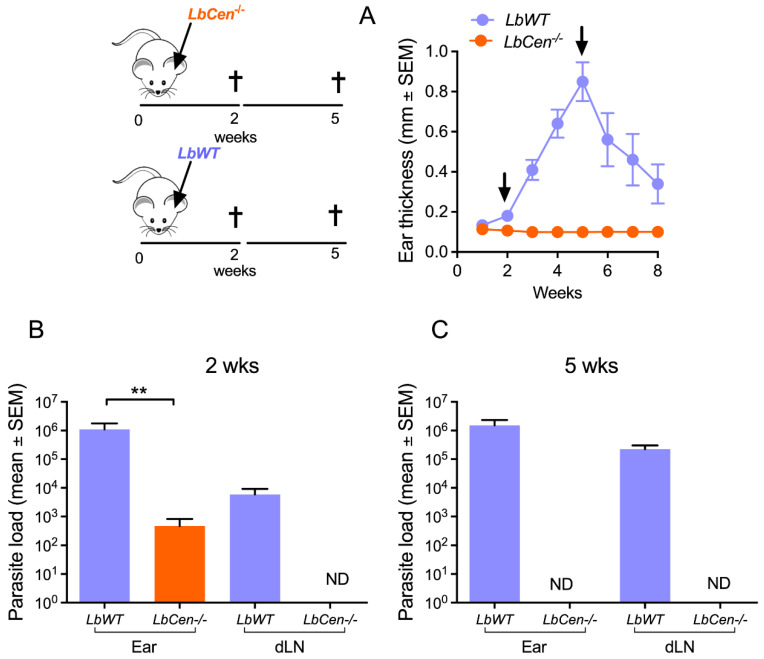
Safety of *LbCen^−/−^* upon experimental infection. BALB/c mice were inoculated with *LbWT* or *LbCen^−/−^* promastigotes. (**A**) Lesion development was measured weekly. Parasite load was measured in dLNs two (**B**) and five (**C**) weeks post challenge. Results are expressed as means ± SEM and are from one representative experiment, performed with five mice. ** *p* < 0.01. ND, not detected.

**Figure 4 vaccines-12-00310-f004:**
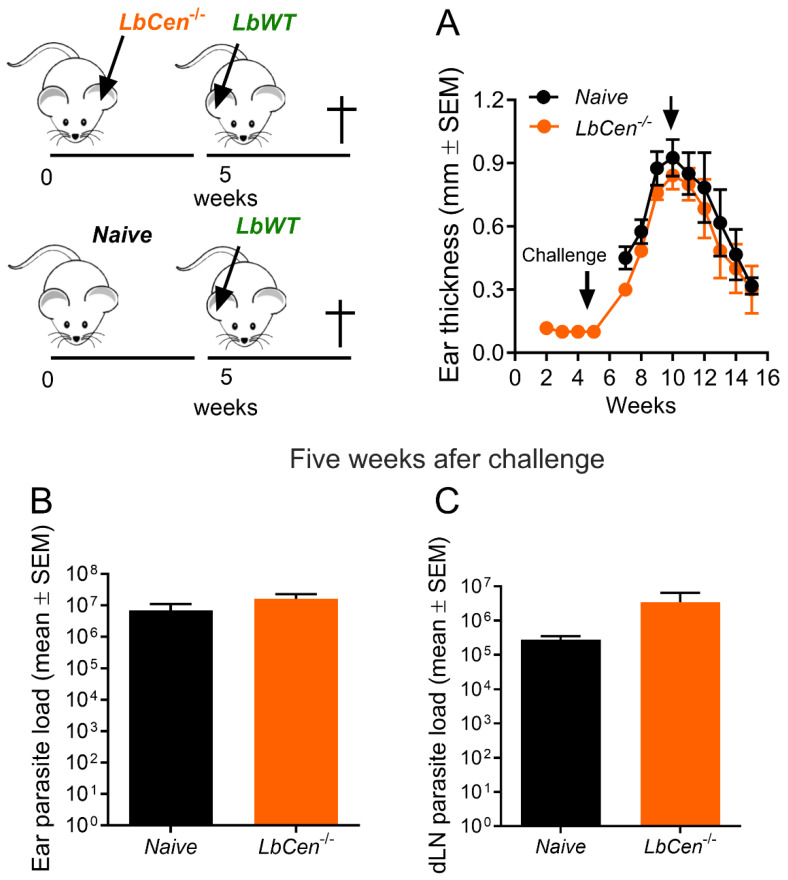
Efficacy of *LbCen^−/−^* against *L. braziliensis*. BALB/c mice were inoculated with *LbCen^−/−^* stationary promastigotes in the ear dermis. Five weeks later, mice were challenged with *LbWT* and (**A**) lesion development was measured weekly. (**B**) Parasite load was measured five weeks after challenge in the ear (**B**) and dLNs (**C**). Results are expressed as means ± SEM and are from one representative experiment performed with six mice.

**Figure 5 vaccines-12-00310-f005:**
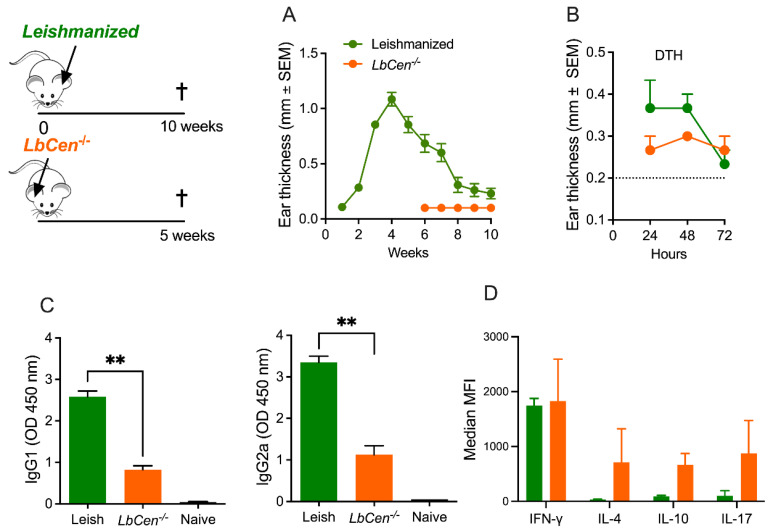
Immune response induced by *LbCen^−/−^* versus a primary *L. braziliensis* infection. BALB/c mice were primary infected with LbWT and left to heal or were inoculated with *LbCen^−/−^* promastigotes. (**A**) Lesion development was measured weekly. (**B**) DTH response to SLA, serology (**C**) IgG1 and IgG2a, and (**D**) cytokine production in dLNs were measured ten weeks after mice healed from the primary infection (*LbWT*) or four weeks after inoculation with *LbCen^−/−^*. Results are expressed as means ± SEM and are from one representative experiment, performed with five mice. ** *p* < 0.01.

**Figure 6 vaccines-12-00310-f006:**
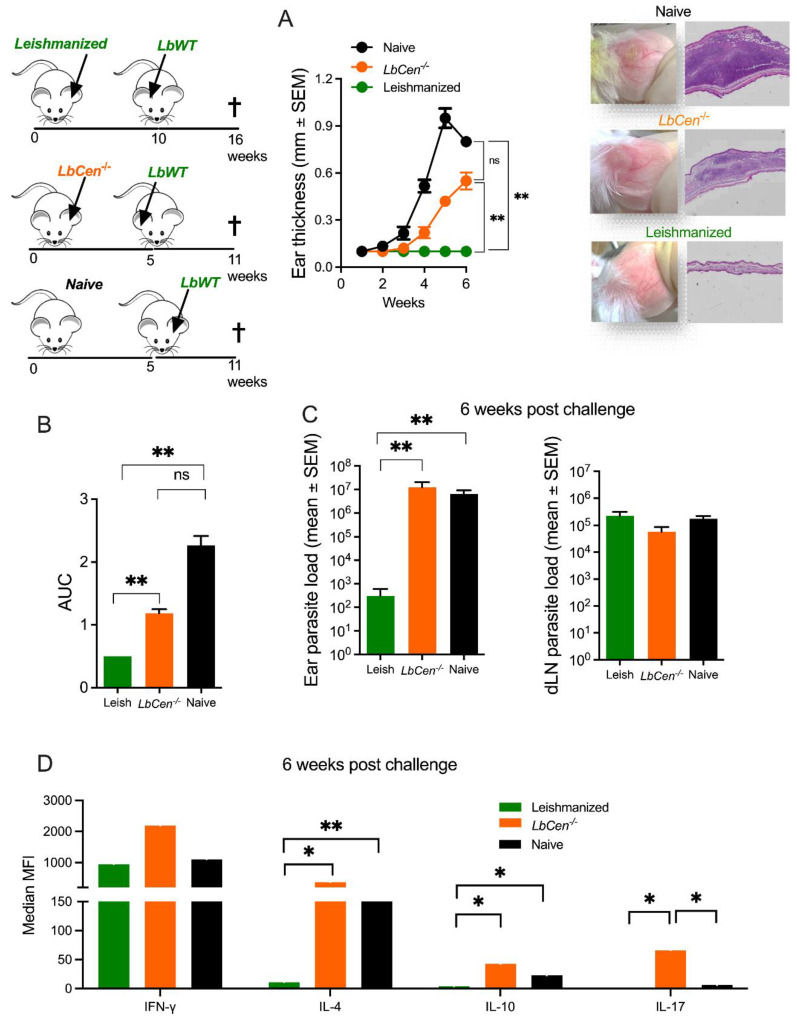
Efficacy of leishmanization vs. immunization with *LbCen^−/−^*. BALB/c mice were inoculated with *LbWT* or *LbCen^−/−^* promastigotes. Healed (*LbWT*) or immunized (*LbCen^−/−^*) mice were challenged with *LbWT*. Controls consisted of age-matched naïve mice. (**A**) Lesion development was measured weekly and histopathological analysis of ears was performed six weeks post challenge. (**B**) Areas under the curves (AUCs) of ear thicknesses shown in (**A**). (**C**) Parasite load in ears and draining lymph nodes six weeks post challenge. (**D**) Cytokine response in draining lymph nodes six weeks post challenge. Results are expressed as means ± SEM and are from one representative experiment, performed with six mice. * *p* < 0.05; ** *p* < 0.01. ns, not significant.

**Figure 7 vaccines-12-00310-f007:**
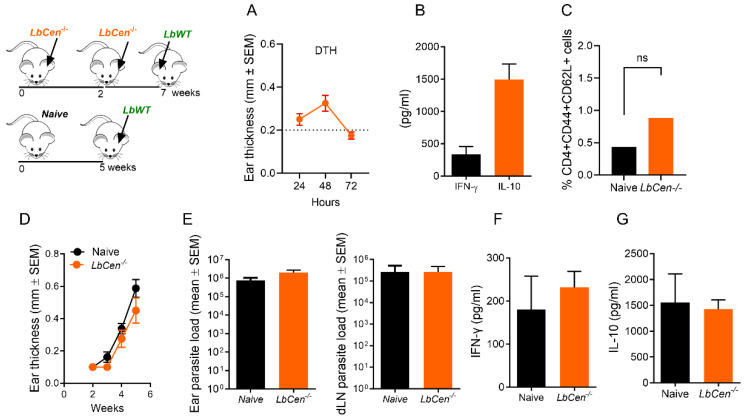
Prime boost with *LbCen^−/−^* cannot improve immunity against *L. braziliensis*. BALB/c mice were inoculated with 3 × 10^6^ *LbCen^−/−^* promastigotes and, two weeks later, mice received a booster inoculation. (**A**) DTH response and (**B**) cytokine response in dLNs cells following re-stimulation with SLA. (**C**) Frequency of CD4^+^ T central memory cells (CD4^+^CD44^+^CD62L^+^). Evaluations were performed two weeks after the boost inoculation. Mice were later challenged with 10^5^ *LbWT* promastigotes and lesion development was measured weekly (**D**). (**E**) Parasite load, (**F**) cytokine production in DLN cells, and frequency of CD4^+^ T central memory cells (**G**) were determined five weeks after challenge. Results are expressed as means ± SEM and are from one representative experiment, performed with eight mice. ns, not significant, dotted line represents baseline ear thickness.

**Figure 8 vaccines-12-00310-f008:**
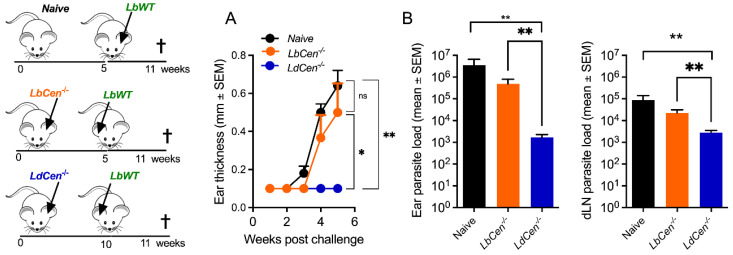
Protective immunity of *LdCen^−/−^* against *L. braziliensis*. Mice were immunized with *LdCen^−/−^* promastigotes and, five weeks later, mice were challenged with *LbWT* promastigotes. Lesion development was monitored weekly (**A**). Parasite load in ears and draining lymph nodes (**B**) was determined by limiting dilution five weeks post challenge. Results are expressed as means ± SEM and are from one experiment, performed with six mice. * *p* < 0.05; ** *p* < 0.01. ns, not significant.

## Data Availability

The data presented in this study are available on request from the corresponding author.

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
