# Peer review of "Immunization with centrin-Deficient Leishmania braziliensis Does Not Protect against Homologous Challenge"

_vaccines, 2024, doi:10.3390/vaccines12030310_

Round 1

Reviewer 1 Report

Comments and Suggestions for Authors

It is important to define what is exactly meant by "lesions"and "inflammation".  It is not clear how centrin-/- L. braziliensis induce higher production of the pro-inflammatory IL-1 production, reflecting a stimulated inflammasome, that do not result in lesions.

Taking the findings together,  the authors should consider that the procedures used for induction of centrin deficiency in L-braziliensis were accompanied by more or less subtle and significant changes in the parasite surface membrane and secreted antigens.  Centrin deficient parasites apparently expose or produce a quite protective antigen, lacking or concealed in wild type counterparts.  It would be interesting in the future to map, compare, contrast, using abstracting and deduction, the transcriptome and/or proteome of these different wild type and centrin-deficient amastigotes.

Author Response

REV1

It is important to define what is exactly meant by "lesions"and "inflammation".  It is not clear how centrin-/- L. braziliensis induce higher production of the pro-inflammatory IL-1 production, reflecting a stimulated inflammasome, that do not result in lesions.

Authors’ response: In our study, "lesions" signify visible tissue damage linked with inflammation, while "inflammation" refers to immune responses involving inflammatory mediators and cells. Our results show that wild-type L. braziliensis supresses IL-1b production in infected macrophages whereas centrin-deficient L. braziliensis does not. There is no increase in IL-1b in macrophages infected with centrin deficient L braziliensis since the levels of IL-1b are  similar to uninfected Macrophage.  

Taking the findings together, the authors should consider that the procedures used for induction of centrin deficiency in L-braziliensis were accompanied by more or less subtle and significant changes in the parasite surface membrane and secreted antigens.  .  It would be interesting in the future to map, compare, contrast, using abstracting and deduction, the transcriptome and/or proteome of these different wild type and centrin-deficient amastigotes.

Authors’ response: We agree with the reviewer’s comment and are currently analyzing proteomic profiles of wild-type and centrin-deficient parasites to understand the mechanisms behind observed phenotypic differences. In future investigations, we will integrate transcriptomic and metabolomic analysis to comprehensively characterize molecular alterations associated with centrin deficiency in L. braziliensis.  as we have previously determined in LmCen-/- and LmexCen-/- parasites.

Reviewer 2 Report

Comments and Suggestions for Authors

Dear authors,

after the revising your manuscript "IMMUNIZATION WITH CENTRIN-DEFICIENT LEISHMANIA BRAZILIENSIS DOES NOT PROTECT AGAINST HOMOLOGOUS CHALLENGEI came to the following conclusion: my suggestion is that your manuscript can be accepted after minor revision. In order to be accepted you have to make some minor technical corrections.

My concerns are:

  • There are some surplus references in the list while not present in the text. You should delete them.
  • Please arrange the references according to the Instructions for authors of the journal
  • Please, read all the comments in the “corrected” version of the manuscript and correct your data.

Also I have to say, you have done an excellent job. My congrats

Best regards 

Author Response

We thank the reviewer for the corrections. Modifications are highlighted in yellow in revised version.

Reviewer 3 Report

Comments and Suggestions for Authors

The paper from Avendano-Rangel seeks to use a centrin-deficient L. braziliensis parasite as a vaccine to protect against wild-type infection. The outcome of the study showed that Lbcen-/- did not protect against subsequent  wt infection, which is surprising as Ldoncen-/- did. Indeed, Ldon cen-/- would appear to be the vaccine strategy to develop, and the current paper presents a cautionary tale. 

Overall, this is a well-written paper, and I have only a few minor comments for clarification - 

1.Lines 79-85 are results in the introduction. Please remove and just end on a hypothesis or aims of your study.

2.Line 117: why do you prime with LPS? It must be made clear why BMDM are primed with LPS.

3. Line 136. ends with humoral response

4. There are a lot of animals used in the study, but it's not made clear in the figures legends a) how many expts. done and b) how many mice used. These need to be clarified in the legends of figures.

5. Figure 2. What is Mo in the figure?

6. Line 311. thar spelling. That.

7. Line 363. The journal may not accept data not shown, so consider putting these in supplementary.

8.  Line 471. LRV (needs a reference). What is LRV - please explain.

9. General point on discussion - I think the authors should explain in more detail the differences between Lb and Ldon. Why does the latter induce cross-protection? Can it be more than just the differentiation between Th1 and Th2 immune response induction? How different are these species?

Comments on the Quality of English Language

English language very god; only a couple of minor typos. You might want to consider breaking up the discussion into a few more paragraphs.

Author Response

REV3

The paper from Avendano-Rangel seeks to use a centrin-deficient L. braziliensis parasite as a vaccine to protect against wild-type infection. The outcome of the study showed that Lbcen-/- did not protect against subsequent  wt infection, which is surprising as Ldoncen-/- did. Indeed, Ldon cen-/- would appear to be the vaccine strategy to develop, and the current paper presents a cautionary tale. 

Overall, this is a well-written paper, and I have only a few minor comments for clarification - 

1.Lines 79-85 are results in the introduction. Please remove and just end on a hypothesis or aims of your study.

Authors’ response: The paragraph was modified as suggested.

2.Line 117: why do you prime with LPS? It must be made clear why BMDM are primed with LPS.

Authors’ response: We wished to evaluate whether LbCen-/- would be able to modulate the macrophage response to lipopolysaccharide (LPS), which stimulates macrophages through TLR4, resulting in the production of inflammatory mediators. In this context, macrophages primed with LPS are prone to produce inflammatory mediators and we evaluate whether a subsequent exposure to leishmnai is able to change this outcome. With the exception of IL-1b production, this was not observed.

  1. Line 136. ends with humoral response

Authors’ response:

  1. There are a lot of animals used in the study, but it's not made clear in the figures legends a) how many expts. done and b) how many mice used. These need to be clarified in the legends of figures.

Authors’ response: The information was added to the figure legends, as suggested.

  1. Figure 2. What is Mo in the figure?

Authors’ response: Mo refer to control cultures, performed with Macrophages alone, not exposed to leishmania. The figure legend was modified to include this information.

  1. Line 311. thar spelling. That.

Authors’ response: Thank you for the correction.

  1. Line 363. The journal may not accept data not shown, so consider putting these in supplementary.

Authors’ response: Thank you for the suggestion.

  1. Line 471. LRV (needs a reference). What is LRV - please explain.

Authors’ response: We apologize for the abbreviation. LRV refers to Leishmania RNA Virus (LRV). It is an RNA virus that can infect leishmania parasites, causing activation of TLR3. A review on the subject was added, as suggested.

  1. General point on discussion - I think the authors should explain in more detail the differences between Lb and Ldon. Why does the latter induce cross-protection? Can it be more than just the differentiation between Th1 and Th2 immune response induction? How different are these species?

Authors’ response: As the reviewer may appreciate, lack of protection in LbCen-/- immunized animals is an unexpected result. We believe, one of the explanations could be rapid clearance of LbCen-/- in the immunized hosts may partly explain the lack of protection, however our data with prime boost schedule did not significantly alter the result. This suggests that a limited replication of a centrin deletion mutant maybe necessary to generate a robust protective immunity as observed in LdonCen-/- and LmCen-/- mutants. Alternatively, the early metabolic immune regulation appears to be fine-tuned in LmCen-/- and LmexCen-/- mutants towards promoting a pro-inflammatory environment that supports the development of Th1 immunity. Therefore, it remains to be seen what are the metabolomic changes in LbCen-/- parasites that may explain their inability to protect against virulent L. brazilensis infection.